# Exploring 1,3-Dioxolane Extraction of Poly(3-hydroxybutyrate) and Poly(3-hydroxybutyrate-co-3-hydroxyvalerate) from *Methylocystis hirsuta* and Mixed Methanotrophic Strain: Effect of Biomass-to-Solvent Ratio and Extraction Time

**DOI:** 10.3390/polym16131910

**Published:** 2024-07-04

**Authors:** Claudia Amabile, Teresa Abate, Simeone Chianese, Dino Musmarra, Raul Muñoz

**Affiliations:** 1Department of Engineering, University of Campania “Luigi Vanvitelli”, Via Roma 29, 81031 Aversa, Italy; claudia.amabile@unicampania.it (C.A.); teresa.abate@unicampania.it (T.A.); simeone.chianese@unicampania.it (S.C.); dino.musmarra@unicampania.it (D.M.); 2Institute of Sustainable Processes, University of Valladolid, Dr. Mergelina, s/n, 47011 Valladolid, Spain

**Keywords:** 1,3-dioxolane solvent poly(3-hydroxybutyrate) extraction, mixed methanotrophic strain, poly(3-hydroxybutyrate-co-3-hydroxyvalerate) extraction, pure methanotrophic strain, sustainable extraction process

## Abstract

The increasing need for biodegradable polymers demands efficient and environmentally friendly extraction methods. In this study, a simple and sustainable method for extracting polyhydroxybutyrate (PHB) and poly(3-hydroxybutyrate-co-3-hydroxyvalerate (PHB-co-HV) from *Methylocystis hirsuta* and a mixed methanotrophic consortium with different biopolymer contents was presented. The extraction of biopolymers with 1,3-dioxolane was initially investigated by varying the biomass-to-solvent ratio (i.e., 1:2 w v^−1^, 1:4 w v^−1^, 1:6 w v^−1^, 1:8 w v^−1^ and 1:10 w v^−1^) and extraction time (6, 8 and 10 h) at the boiling point of the solvent and atmospheric pressure. Based on the results of the preliminary tests, and only for the most efficient biomass-to-solvent ratio, the extraction kinetics were also studied over a time interval ranging from 30 min to 6 h. For *Methylocystis hirsuta*, the investigation of the extraction time showed that the maximum extraction was reached after 30 min, with recovery yields of 87% and 75% and purities of 98.7% and 94% for PHB and PHB-co-HV, respectively. Similarly, the extraction of PHB and PHB-co-HV from a mixed methanotrophic strain yielded 88% w w^−1^ and 70% w w^−1^ recovery, respectively, with 98% w w^−1^ purity, at a biomass-to-solvent ratio of 6 in 30 min.

## 1. Introduction

Plastic consumption and uncontrolled waste disposal have dramatically increased in recent decades, reaching 460 million tonnes produced in 2019, according to the Organisation for Economic Co-operation and Development (OECD) [1]. Unfortunately, a significant portion of this plastic cannot be recycled, leading to pollution in landfills and the natural environment. Given the above, urgent solutions are required to maintain the functional advantages of plastics while protecting natural ecosystems from destruction. In this context, bioplastics are a more sustainable alternative to fossil-based plastics already in the market, but at higher costs than their fossil counterparts. PHAs are thermoplastic biopolymers that have been in the spotlight for several decades due to their outstanding properties and are regarded as a promising alternative to conventional plastics. PHAs are polymers that, under stress conditions [2], can be accumulated in the cytoplasm of a wide range of bacterial species [3] as intracellular carbon and energy reservoirs. PHB was the first PHA to be discovered in 1926 by Lemoigne. These biodegradable bioplastics can be used in many different sectors, such as medicine, pharmaceuticals and packaging [4,5,6]. PHB is a biodegradable, biocompatible, bio-based and non-toxic polymer with physical, chemical and mechanical properties similar to those of polyethylene (PE) and polypropylene (PP) [7]. However, the PHB obtained to date is still of a lower quality than its fossil-based counterparts, limiting its widespread implementation. In this regard, the presence of hydroxyvalerate (HV) in PHB-based polymers entails a significant improvement in its properties. In fact, despite the already known physical–mechanical similarities between PHB and PP, a comparison with PHB-co-HV showed a further reduction in the performance gap with plastics of fossil origin, which could even make it possible to replace them [7]. In particular, compared to PHB-co-HV, PHB is stiffer and more brittle, is less resistant to solvents, and has a lower elongation at break but exhibits similar tensile strengths [7]. In addition, both crystallinity and melting temperature decrease linearly with HV content [8]. The decomposition temperature is above 200 °C in all cases, making the copolymer more ductile, flexible and processable [8]. The inclusion of the HV fraction depends on the availability of expensive precursors such as valeric acid; in general, the higher the concentration of the precursor, the higher the HV fraction produced. However, increasing the HV fraction above a certain value could increase costs without improving the performance and functionality of PHAs [7].

The main factors limiting the large-scale production of PHAs and the effective replacement of fossil-based plastics are their high costs [9] and current unsustainable production systems [10] associated with recovering the accumulated polymer. The enrichment of PHB with hydroxyvalerate can improve biopolymer quality, but the cost and implementation of an effective, economical and sustainable extraction system remain challenging. Several techniques, such as chemical, physical and enzymatic methods [11] or a combination of them, are reported in the literature [12]. However, conventional techniques have limitations in terms of operating costs, degradation and the quality of the final polymer or the amount of endotoxins produced [13]. The most accepted and implemented technique today is solvent extraction because it guarantees a high yield of recovery and purity, low levels of endotoxins and a good-quality polymer, although it has high operating costs, even if solvent recovery can reduce the cost of the process [14,15]. It is sufficient to consider the percentage reduction in the fresh solvent required for the same process, as reported by Abate et al. [16]. This method is based on the high solubility of the polymer in some solvents, promoting cell rupture, followed by the addition of an anti-solvent in which the polymer is not soluble. Subsequently, the solvent is easily recovered by liquid–solid separation or evaporation.

Despite the advantages of solvent extraction, the main limitation of this technique is the use of large volumes of toxic solvents such as chloroform and dichloromethane to avoid the formation of highly viscous solutions [17]. In some cases, the application of a pretreatment could be required to improve the performance of the process, thus contributing to an additional cost [18].

In this context, a few green solvents, such as butyl acetate [19], ethylene carbonate [20] and 1,3-dioxolane [21,22], have been tested in recent years to make the process and the final polymer truly sustainable. The final recovery yields obtained with these green solvents were comparable to those obtained with chloroform (around 98%), confirming the feasibility of replacing chloroform, even at a large scale. In particular, PHB recovery of 96% for butyl acetate, ≈99% for ethylene carbonate and 93% for 1,3-dioxolane, with a purity of 98% in all cases, has been reported. On the other hand, very few solvents have been tested for PHB-co-HV [17]. Among the solvents tested to date, 1,3-dioxolane is the least efficient in terms of final process yield, but it has many advantages, including a good-quality product, compatibility with water, affordable prices, easy availability and safety, and the ability to support effective separation of the polymer obtained by precipitation. Moreover, it has been reported to be non-toxic, odorless, easy to evaporate and environmentally friendly [22].

Thus, 1,3-dioxolane, which is less dense than water, remains in the upper fraction (with the recovered polymer). At the same time, the residue settles in the bottom aqueous fraction (contrary to what happens with chloroform), facilitating its subsequent recovery [21].

Chloroform is currently the best extraction solvent because it is cheap and efficient [23], but it is toxic and hazardous to humans and the environment [24,25], which limits its large-scale development [26]. Therefore, the use of chloroform provides a biopolymer of reasonable quality but through an unsustainable process. On the other hand, the use of green solvents, such as 1,3-dioxolane, allows for the development of a process that is simple, economical and as efficient as conventional solvents but uses renewable resources and environmentally friendly solvents that are not carcinogenic, toxic, explosive or self-igniting [15,21]. Although the cost of chloroform is lower, the entire extraction process, including equipment and energy costs, contributes to determining the selling price of PHB and PHB-co-HV at 5–6 USD/kg [27], which represents more than 50% of the total cost of the process [27]. Furthermore, the precipitation phase is simple and fast due to the complete miscibility of 1,3-dioxolane in water. Thus, the biopolymer, which is hydrophobic, precipitates in a few seconds and is easily recovered because it is on the surface, facilitating the downstream extraction process [21]. In addition to the advantages already mentioned, this study also demonstrates the greater simplicity of the recovery phase. Therefore, using 1,3-dioxolane can promote an environmentally friendly and cost-effective extraction process.

This study aimed to investigate the effects of extraction time and the biomass-to-solvent ratio on the recovery and purity of the biopolymer extracted with different biomass and biopolymer types and contents. The influence of five biomass-to-solvent ratios (w v^−1^) at three different extraction times was tested, keeping the temperature constant. For all experiments, water was used as an anti-solvent. Under the best extraction conditions in terms of biomass-to-solvent ratio and time (based on the experimental evidence of recovery and purity), the kinetics of biopolymer extraction from *M. hirsuta* and a mixed methanotrophic culture were investigated. To the best of the authors’ knowledge, this study represents the first attempt to develop a unified extraction protocol for the sustainable recovery of PHB and PHB-co-HV, involving consolidated pure strains and mixed methanotrophic strains. Compared to previous studies on PHA extraction, this study introduces several innovations, targeting polymers with varying HV contents and evaluating the effect of the extraction time of the proposed process to determine the minimum time necessary to achieve high-recovery and -purity yields.

## 2. Materials and Methods

### 2.1. Chemicals, NMS Medium and Biopolymer Production

#### 2.1.1. Chemicals

Two PHA-accumulating bacterial cultures were used: a pure culture, *Methylocystis hirsuta*, and a mixed methanotrophic culture. The methanotrophic strain *Methylocystis hirsuta* CSC1 was purchased from the Leibniz Institute DSMZ (Berlin, Germany). The mixed methanotrophic culture was enriched in-house by combining the indigenous microbial community from Sphagnum and activated sludge from the Valladolid wastewater treatment plant [28]. In the latter case, the same mineral medium was used for the growth and accumulation phases, but the medium was deprived of the nitrogen source.

The salts required for mineral salt medium preparation consisting of Na_2_MoO_4_·2H_2_O, CoCl_2_, FeSO_4_·7H_2_O, FeEDTA, NiCl_2_·6H_2_O, ZnSO_4_·7H_2_O, Na_2_HPO_4_·12H_2_O and H_3_BO_3_ were purchased from Sigma Aldrich (St. Louis, MO, USA), KNO_3_ from Labkem (Barcelona, Spain) and CaCl_2_·2H_2_O, MgSO_4_·7H_2_O, Na_2_EDTA·2H_2_O, CuSO_4_·5H_2_O, MnCl_2_·4H_2_O and KH_2_PO_4_ from PanReac AppliChem (Barcelona, Spain). Valeric acid (≥99%), purchased from Sigma Aldrich, was added to the nitrogen-free medium to produce PHB-co-HV. The 1,3-dioxolane (99%) used for the extraction of the polymer was purchased from Sigma Aldrich (St. Louis, MO, USA), as well as all chemicals required for the gas chromatographic measurement: chloroform (≥99.8%), 1-propanol (99.7%), hydrochloric acid (37% w v^−1^), benzoic acid (≥99.5%) and PHB-co-HV with 12%mol HV (≥99.99%). The oxygen (99.5%) and methane (99.95%) required to feed the methanotrophic cultures during both the growth and accumulation phases were purchased from Abelló Linde S.A. (Barcelona, Spain) and Carburos Metalicos (Barcelona, Spain), respectively.

#### 2.1.2. Culture Medium

Both bacterial strains were grown on an MSM containing macronutrients (g L^−1^, 0.2 CaCl_2_·2H_2_O, 1.0 KNO_3_) and micronutrients, the latter supplied by adding 1 mL of a trace-element solution (mg L^−1^, 0.38 Fe-EDTA, 0.4 Na_2_MoO_4_·2H_2_O, 0.3 Na_2_EDTA·2H_2_O, 1.0 CuSO_4_·5H_2_O, 0.5 FeSO_4_·7H_2_O, 0.4 ZnSO_4_·7H_2_O, 0.015 H_3_BO_3_ 0.01, 0.03 CoCl_2_, 0.02 MnCl_2_·4H_2_O, NiCl_2_·6H_2_O). The pH was adjusted to 6.8 by adding 10 mL of a buffer solution containing 72 g L^−1^ of Na_2_HPO_4_·12H_2_O and 26 g L^−1^ of KH_2_PO4. The medium was autoclaved at 121 °C for 30 min in a Raypa steam sterilizer (Barcelona, Spain) before use.

#### 2.1.3. Biopolymer Production

Each strain was inoculated in 25 mL of the MSM (10% v v^−1^) in 125 mL serum bottles, which were sealed tightly to prevent gas leakage. The headspace of all bottles was then filled with a mixture of oxygen and methane (2:1 v v^−1^). More specifically, a stream of filtered oxygen (0.22 μm; Millex GP, Merck, Madrid, Spain) was introduced for approximately 5 min, and then, 25 mL of the oxygen headspace atmosphere was replaced with the same amount of methane using a gastight syringe (Hamilton 1050 TLL, Reno, NV, USA). Finally, the serum bottles were incubated in an orbital shaker at 200 rpm and 25 °C for 6 days. The headspace of the cultures was replaced when the methane was depleted. Then, 10 mL (2% v v^−1^) of these active cultures was inoculated into 2.15 L bottles containing 500 mL of the MSM, previously autoclaved, and incubated under an O_2_/CH_4_ atmosphere of 66.7:33.3% v v^−1^. Cultures were grown under continuous stirring at 350 rpm (Variomag, Thermo Fisher Scientific, Bartlesville, NO, USA) and at room temperature (25 °C) for approximately one week. The same protocol was used to activate and grow the two bacterial cultures, but sterile conditions were only maintained during the cultivation of *M. hirsuta* [29].

At the end of the growth phase, both cultures were resuspended in a N-free medium for the accumulation phase. Valeric acid (VA) (5 mL of a 3.7% v v^−1^ VA solution) was only added when the accumulation of PHB-co-HV was desired. In this regard, López et al. [30] reported that the presence of valeric acid at ~30% of the carbon supplied does not inhibit *M. hirsuta* growth but favors the accumulation of the copolymer in the biomass. Overall, 25 mL (5% v v^−1^) of the inoculum containing the biomass grown in the previous phase was resuspended in 500 mL of nitrogen-deprived sterile medium to accumulate PHB and PHB-co-HV. The headspace was filled with a mixture of oxygen and methane as described above, and the cultures were incubated at 350 rpm and 25 °C for 2–3 days (corresponding to maximum accumulation), as described above.

After the accumulation phase of PHB and PHB-co-HV, for both bacterial cultures, the same biopolymer pre-quantification procedure was applied: the total volume of culture broth was divided equally into falcon tubes containing aliquots of 40 mL and then centrifuged at 10,000 rpm for 10 min using a Fisher Bioblock Scientific 2-16P (FisherScientific, Madrid, Spain). The pellets were then resuspended in 2 mL of nitrogen-free medium, transferred into a 2 mL Eppendorf tube and centrifuged at 10,000 rpm for 10 min (Spectrafuge 24D, Labnet International, Inc., Madrid, Spain). Finally, the supernatant was discarded, and the pellet obtained was stored at −4 °C (Figure 1). All samples preserved were used to apply the extraction procedure with 1,3-dioxolane (each measurement was carried out in triplicate). Two samples were used to determine the total suspended solid (TSS) concentrations, and two additional samples were used to quantify the accumulated PHB/PHB-co-HV through standard extraction with chloroform, as described in the Analytical Procedures Section.

### 2.2. PHB and PHB-co-HV Extraction with 1,3-Dioxolane

The biomass from the biopolymer accumulation phase (*M. hirsuta* or the mixed methanotrophic strain) was used during the extraction procedure. The extraction of PHB and PHB-co-HV was carried out by varying the biomass-to-solvent ratio, Bio/Sol (w v^−1^), and the extraction time. Note that this ratio is calculated based on the initial amount of biomass containing PHA granules, while the non-polymeric cellular material (NPCM) is the solid residual fraction removed at the end of the process. Frozen biomass samples, i.e., 10.2 mg of biomass containing PHB and 8 mg of biomass containing PHB-co-HV (on a dry basis), were used for extraction. First, samples were thawed overnight at 4 °C; then, 1,3-dioxolane was added to the frozen biomass to reach biomass-to-solvent ratios (Bio/Sol) (w v^−1^) of 2, 4, 6, 8 and 10, which corresponded to 5.10, 2.55, 1.70, 1.28 and 1 mL of solvent for the biomass containing PHB, and 4, 2, 1.3, 1 and 0.8 mL of solvent for the biomass containing PHB-co-HV. First, the appropriate volume of solvent was added to each sample and mixed until the biomass was completely dissolved. Avoiding the biomass remaining in aggregate form was essential, as this could complicate the subsequent extraction phase. The prepared samples were inserted into a thermoreactor at 74 °C for 6, 8 and 10 h with vortex agitation every 30 min. Extraction was performed at atmospheric pressure. Considering the linear relationship between solubility and temperature reported in the literature [31], to favor the dissolution and extraction of the biopolymer, the highest achievable temperature, i.e., the boiling point of the solvent at atmospheric pressure (T_eb,DIOX_ = 76 °C), was chosen [21]. Under these conditions, it is obvious that the solvent is highly volatile. However, as it has not yet reached its boiling point, and thanks to the use of appropriate equipment (the tests were carried out in airtight glass tubes sealed with screw caps), no significant loss of solvent was observed. As a further check, the solvent level inside the container was marked at the beginning of the test and compared with the level at the end of the experiment.

After extraction, the dissolved polymer was recovered through precipitation by adding 3 volumes of MilliQ water. The lower density of 1,3-dioxolane compared to that of water facilitated the recovery of PHA from the liquid fraction, which was then washed in MilliQ water and dried on filter paper overnight at room temperature. The effective amount of the recovered polymer was assessed by weighing it after drying. The extraction performance was evaluated in terms of recovery (R) and purity (P).

The recovery was calculated from duplicate samples. For the calculation of purity, for each biomass-to-solvent ratio, all samples were pooled according to the extraction time (6, 8 and 10 h) and the average value was assessed.

### 2.3. Extraction Kinetics of PHB and PHB-co-HV

The extraction kinetic evaluation was designed based on the results obtained from the tests of the PHA extraction process, using a biomass-to-solvent ratio of 6 and an extraction time of 6 h as the upper limit. PHA recovery was assessed at 30 min, 1 h, 1.5 h, 2 h, 3 h, 4 h, 5 h and 6 h, while biopolymer purity was estimated at 30 min, 2 h, 4 h and 6 h. The recovery and the purity were calculated in triplicate, but only the average values were reported.

### 2.4. Recovery and Purity Assessment

Conventional chloroform extraction was used as a benchmark for recovery and purity. In line with the scientific literature, it was assumed that the highest recovery yield of PHAs from biomass was achieved using the standard method [15,21,32]. Accordingly, the final recovery and purity were evaluated in terms of the amount of biopolymer extracted with chloroform.

The final recovery of PHB and PHB-co-HV was calculated according to Equation (1). In particular, PHAs_extr,DIOX_ was measured as the weight of the dry polymer recovered after the precipitation phase; PHAs_extr,CHCl3_ was calculated as the product between the concentration of the polymer extracted by the standard method with chloroform, measured by gas chromatography, and the initial volume of the sample. The extractions were performed using the same amount of dry biomass (10.2 mg for PHB extraction and 8 mg for PHB-co-HV extraction).
(1)$$R\;[\%]=\frac{PHAs_{extr,DIOX}[mg]}{PHAs_{extr,CHCl3}[mg]}\times 100$$

The purity of PHB and PHB-co-HV was calculated according to Equation (2):(2)$$P\;[\%]=\frac{PH{As}_{GC,DIOX}[mg]}{PHAs_{extr,DIOX}[mg]}\times 100$$
where PHAs_GC,DIOX_ is the amount of biopolymer precipitated after extraction with 1,3-dioxolane, resuspended in chloroform and re-extracted by the standard chloroform extraction method.

### 2.5. Analytical Procedures

In order to estimate the recovery yield and the purity of PHB and PHB-co-HV from *M. hirsuta* and mixed methanotrophic strains, the biopolymer measurement was performed by using the method of [33], modified as reported by [34]. More specifically, to quantify %PHAs_CHCl3_ and its purity, 1 mL of a mixture of 1-propanol and hydrochloric acid (80:20 v v^−1^) was added to the pellet. At this point, the mixture was agitated by vortexing and 2 mL of chloroform (extraction solvent) and 10 µL of benzoic acid (5 g L^−1^) (internal standard solution) were added. The samples were then incubated at 100 °C for 4 h. At the end of the digestion period, the samples were allowed to cool to room temperature, 1 mL of MilliQ water was added and the samples were stirred again to promote the separation of the organic and inorganic phases. Finally, the organic phase was collected and filtered (0.22 μm; Merck). Measurements were performed in an Agilent 7820A GC coupled to a 5977E MSD (Agilent Technologies, Santa Clara, CA, USA) and equipped with a DB-WAX column (Agilent Technologies, Santa Clara, CA, USA) using benzoic acid and PHB-co-HV (12% mol HV) as internal and external standards, respectively. Finally, biomass concentration, expressed as TSS, was estimated according to Standard Methods [35].

## 3. Results

### 3.1. PHB Extraction with 1,3-Dioxolane from M. hirsuta

A culture of *M. hirsuta* containing around 62% w w^−1^ of PHB was used for this test. During the extraction with 1,3-dioxolane, thanks to the solvent’s properties, the precipitated solid remained in the upper fraction of the solvent/anti-solvent mixture. The subsequent addition of water caused the formation of a swollen white polymer. This phenomenon was previously observed during the extraction of PHAs with 1,3-dioxolane and confirmed in this study [15]. The solid was washed with MilliQ water and left overnight on filter paper at room temperature. The biopolymer precipitated and dried is shown in Figure 2a and Figure 2b, respectively.

Through this chemical phenomenon, once the solid polymer is removed, 1,3-dioxolane can be separated from water through fractional distillation, as there is a significant temperature difference (>20 °C) in the boiling points between water and 1,3-dioxolane. This not only avoids the production of large quantities of waste but also reduces the final cost of the process in terms of raw material purchase. Due to the small volumes used in this study, solvent recirculation was not feasible. Nevertheless, this operation is certainly feasible on a large scale. However, it is impossible to reuse 100% of the solvent to avoid the accumulation of inert compounds in the streams; so, it is more practical to waste at least 1–2% and replace it with fresh solvent [14]. Ideally, approximately 98–99% of 1,3-dioxolane should be recovered and recirculated into the extraction process [14]. The solvent leaving the plant (1–2%) can be further recovered from mixtures or solutions via purification and reused [36].

According to Figure 3, for all Bio/Solv ratios (w v^−1^), the recovery of PHB from *M. hirsuta* biomass increased with contact time, reaching a maximum value of ≈96% w w^−1^ at ratios of 4, 6, 8 and 10 at a time of 10 h. This behavior was consistent with previous findings, which demonstrated that extending the time cycle increases extraction yields [37]. Lower but similar results were obtained at 6 and 8 h, corresponding, respectively, to ≈87 ± 5.9% w w^−1^ and 91 ± 5.1% w w^−1^ at a biomass-to-solvent ratio of 4 and 89 ± 4.2% w w^−1^ and 94 ± 1.3% w w^−1^ at a ratio of 6. These results were similar to those reported for sodium hypochlorite, ethylene carbonate and chloroform, with values of 86, 98.6 and 99%, respectively, but were higher when compared to other green solvents such as dimethyl carbonate, acetone or ethanol, with values ranging from 70 to 80% [38]. Therefore, the biomass-to-solvent ratio was set at 6, based on the empirical solvent consumption and final recovery yields herein obtained. In this context, the extraction time determines the sizing of the plant and its CAPEX, which ultimately governs its widespread implementation at an industrial scale.

Figure 4 shows purity yields of ≈67 ± 4% w w^−1^, 45 ± 2.8% w w^−1^ and 44 ± 2.9% w w^−1^ in the polymers extracted during 6 h, 8 h and 10 h, respectively. Interestingly, despite the lower recovery, a greater proportion of the accumulated PHB was obtained after 6 h of solvent extraction as a result of a higher purity. Hence, balancing the recovery and purity yields, the PHB extraction efficiency at 6 h resulted in being comparable to that recorded at 10 h. In view of the results obtained in terms of purity, the maximum contact time was set at 6 h. Additional tests were carried out to investigate the effect of extraction time during PHA recovery with 1,3-dioxolane since, from an industrial point of view, the shorter the extraction time, the lower the energy consumption and the CAPEX of the process.

The results demonstrate the feasibility of this protocol for biomass-to-solvent ratios ranging from 1:2 (w v^−1^) to 1:10 (w v^−1^). The selection of the lower limit (1:2) was based on the amount of solvent needed to completely cover the pellet to be extracted. If the solvent could not cover the entire pellet, the small contact area would limit cell disruption and, therefore, dissolution of the intracellular polymer in the solvent. On the other hand, the choice of the upper limit (1:10) was based on economic considerations since working with too large amounts of solvent is not economically sustainable at an industrial scale.

Previous studies in the literature have successfully worked with smaller solvent volumes but using bacterial biomass with higher PHB contents (approximately 69% w w^−1^), more elaborate extraction protocols and higher operating temperatures (up to 80–100 °C, above the boiling point of 1,3-dioxolane) [15,21,37,39]. Extraction at high temperatures requires greater energy consumption and equipment designed to operate under high pressure, thus increasing the final process cost [20]. Additionally, some studies suggest that there is a maximum extraction temperature beyond which PHA may be damaged [40]. It has been shown that temperatures above 80 °C damage the extracted polymer and reduce the final process yield [21]. Lafferty and Heinzle [41] have also demonstrated that very high temperatures can reduce the molecular weight of the extracted polymer. Yabueng and Napathorn [21] have also tested 1,3-dioxolane extraction at room temperature to reduce energy costs, achieving a recovery of 89% and a purity of 91%. This confirms that the amorphous polymer inside wet cells is likely more soluble than the pretreated and crystallized polymer, as indicated in this study. However, they highlighted that under these conditions, in addition to the increased costs due to a prolonged extraction time (36 h), room temperature may facilitate the activity of the intracellular PHA depolymerase enzyme, which could degrade PHB polymers during prolonged extraction. Indeed, the results confirm a decrease in molecular weight, Young’s modulus and tensile strength compared to those obtained at 80 °C after 6 h of contact. Therefore, it is worth noting that in the present study, the extraction temperature, corresponding to 74 °C, was lower than that used in other reference studies but still ensured high recovery yields and purity.

#### Kinetics of PHB Extraction from *M. hirsuta*

The kinetic tests were carried out to deepen the assessment of the effect of time (30 min–6 h) on PHB recovery and purity at a biomass-to-solvent ratio of 6. Figure 5 shows that the highest recovery (98 ± 1.9% w w^−1^) was achieved after only 3 h of extraction. However, the maximum purity (≈99 ± 0.9% w w^−1^) was achieved after only 30 min, when 87 ± 1% w w^−1^ of the accumulated PHB was extracted. These results are comparable to those obtained in previous studies using chloroform, with recovery yields ranging from 78 to 94% and purities from 86 to 99% [38]. From the results obtained, it emerges that in addition to the reduced temperature, there is also the possibility of completing the entire extraction phase in just 30 min. Such a short contact time can only positively impact the final process costs, especially compared to the 5 and 6 h expected by Yabueng and Napathorn [21] and Wongmoon and Napathorn [15], respectively.

After the peak, the purity of PHB was reduced to about 78 ± 2% w w^−1^, 69 ± 1.9% w w^−1^ and 67 ± 1% w w^−1^ after 2, 4 and 6 h of extraction, respectively, as a result of the extraction of polar bio-compounds present in *M. hirsuta* cells. The reduction in the purity by increasing the extraction time was probably due to an increase in the extraction of impurities (such as lipids and proteins), which were then recovered together with the polymer. It has been shown that the presence of other components or impurities in the matrix can affect extraction efficiency. Some solvents can also extract unwanted components, so it is important to consider cross-solubility to achieve effective separation [31,42,43]. Other studies in the literature also confirmed this trend: Yabueng and Napathorn [21] reported a decrease in purity from 94% w w^−1^ to 77% w w^−1^ when the extraction time was increased from 4 to 36 h, while Mohammadi et al. [43] did not observe any significant decrease in purity when the incubation time was increased from 1 to 5 h, with a purity of 69% w w^−1^ and 65% w w^−1^, respectively. In addition, the physical and chemical properties of the polymer may deteriorate by increasing the extraction time [40].

According to the experimental findings, to balance the recovery and purity yields, the most favorable condition for PHB extraction consists of a biomass-to-solvent ratio of 6 and an extraction time of 30 min, allowing us to obtain 87 ± 1% w w^−1^ recovery and 99 ± 0.9% w w^−1^ purity.

### 3.2. PHB-co-HV Extraction with 1,3-Dioxolane from M. hirsuta

*M. hirsuta* produced PHB-co-HV when valeric acid was supplied during the accumulation phase (5 mL of a 3.7% v v^−1^ VA solution) under nitrogen-limited conditions. More specifically, the cells used contained 60% w w^−1^ of PHAs, with ≈77% w w^−1^ being PHB and the remaining ≈23% w w^−1^ being PHV. The investigation of PHB-co-HV extraction was carried out under the same conditions described for PHB in Section 3.1.

The maximum recovery yield was obtained at a ratio of 6 and leveled off at ratios of 8 and 10 (Figure 6). In particular, the recovery yield increased with the increasing extraction time at higher biomass-to-solvent ratios (1:2 w v^−1^, 1:4 w v^−1^ and 1:6 w v^−1^) but was almost constant at ratios of 8 and 10. Then, when the biomass-to-solvent ratio was fixed at 6, the purity was higher at 6 h (70 ± 4% w w^−1^) than at 8 h and 10 h (52 ± 4% w w^−1^ and 56 ± 2% w w^−1^, respectively) (Figure 7).

The results obtained during the extraction of PHB-co-HV matched those obtained during PHB extraction, where a biomass-to-solvent ratio of 6 was identified as the ideal condition for maximizing the recovery and purity of PHB-co-HV while minimizing the volume of solvent added. Similarly, a lower but purer amount of polymer was recovered after 6 h of extraction, giving a comparable amount of biopolymer to that obtained after 8 and 10 h of extraction. To date, no other studies have been conducted on extracting PHB-co-HV using a similar protocol, making it challenging to compare this study with the existing literature. However, the results confirm the feasibility of using the same extraction technique to recover PHB-co-HV. Nonetheless, they also underscore the need for further investigation of the process.

#### Kinetics of PHB-co-HV Extraction from *M. hirsuta*

After the first hour of extraction of PHB-co-HV with 1,3-dioxolane at a biomass-to-solvent ratio of 6, the recovery yield reached a maximum of 98 ± 2% w w^−1^ and remained roughly constant until the end of the experiment, with a slight decrease at 5 and 6 h (95 ± 3% w w^−1^), as shown in Figure 8. However, the purity peaked after 30 min and then decreased to a constant value of about 70 ± 4.5% w w^−1^ at 2, 4 and 6 h (Figure 8). Here, the recovery peaked almost immediately and then stabilized, unlike the results obtained for PHB, where the maximum recovery was reached after 3 h and decreased by 8% in the following 3 h. This difference can be explained by considering the solubility of the investigated polymers. Narasimhan and co-workers reported that PHB-co-HV is generally more soluble in a wider range of solvents than its PHB homopolymer [44]. Regardless, the experimental findings highlight the same most favorable conditions (Bio/Sol = 6; extraction time = 30 min) for PHB-co-HV extraction with respect to PHB.

### 3.3. PHB and PHB-co-HV Extraction from a Mixed Methanotrophic Culture

The extraction tests of PHB (48% w w^−1^) and PHB-co-HV (35% w w^−1^, with a PHV content of 56% w w^−1^) produced by a mixed methanotrophic strain were performed using a biomass-to-solvent ratio of 6. In line with previous findings, biopolymer recovery increased within the first few hours of extraction, reaching a peak after 4 h (99 ± 1.9% w w^−1^ for PHB and 95 ± 3% w w^−1^ for PHB-co-HV) and then declining slightly to 86 ± 3.1 w w^−1^ and 88 ± 3.9% w w^−1^ for PHB and PHB-co-HV, respectively (Figure 9). Contrary to the observations with *M. hirsuta*, the recovery yield of PHB from the mixed methanotrophic culture was higher than the yield of its copolymer. However, in this case, the difference between the two polymers could be due to the initial content, which was higher for PHB. In both cases, as the contact time increases (at 5–6 h), the recovery yield of PHB is higher than that of PHB-co-HV. On the other hand, the purity of PHB-co-HV was higher, with a peak already after the first 30 min of extraction (≈97 ± 2.2% w w^−1^) and a progressive decrease to 67 ± 3.3%, while the purity of PHB decreased from 96 ± 2% w w^−1^ after 30 min to 38 ± 3% w w^−1^ after 6 h of extraction. This difference in behavior can be related to the use of different bacterial biomasses or the higher solubility of the copolymer in 1,3-dioxolane at longer contact times compared to PHB. However, the highest purity for both polymers was reached after 30 min despite the significant difference in recovery. In line with the results obtained previously with *M. hirsuta*, the trend changes with increasing extraction times (>5 h).

This experiment confirmed that the biopolymers grown and accumulated in mixed methanotrophic cultures can also be extracted with 1,3-dioxolane. The recovery and purity yields herein obtained agree with those reported in the literature [21,22], but a higher biomass-to-solvent ratio and shorter extraction times were used in this study. In contrast to PHB, the extraction of its copolymer PHB-co-HV has been significantly less investigated. Compared to research studies evaluating the extraction of the copolymer, a lower recovery yield but with a higher purity was obtained. The difference could probably be due to the use of a mixed strain rather than a pure strain or to the significantly different share of HV present in the polymer accumulated in the model biomass: more specifically, 10% w w^−1^ in the study by Yang et al. [17] and 22.6% w w^−1^ and 56.86% w w^−1^ in this study. The results obtained in this study pave the way for the use of mixed strains, which are still relatively unexplored and underutilized. Despite yielding lower outputs, replacing pure strains facilitates the growth and recovery process, as these bacteria do not require sterile conditions. This makes the process faster and more cost-effective.

### 3.4. Comparison of Extraction Yields for Different Contents of PHB and PHB-co-HV

A comparison of the recovery and purity yields based on the initial polymer content has been conducted. Table 1 indicates that the initial content of PHB and PHB-co-HV accumulated in the pure culture is higher than that produced by the mixed consortium. According to a previous study, the recovery of PHB-co-HV was lower for both cultures compared to PHB, likely due to its higher molecular mass [39]. Under consistent conditions, which mark the highest purity obtained (Bio/Sol = 6, t = 30 min), the recovery (R) and purity (P) yields of PHB were unaffected by the initial biomass content. In this case, the mixed consortium resulted in slightly higher recovery and lower purity. Conversely, PHB-co-HV extraction from *M. hirsuta* showed higher recovery yields (75% vs. 70%) and lower purity (94% vs. 97%) compared to the mixed consortium. The more pronounced difference in recovery yields could be attributed to the greater disparity in polymer content stored by the two cultures. For PHB-co-HV, there is a nearly 30% difference in total polymer content, which is twice the difference observed for PHB (14%). Overall, the results suggest that with the proposed extraction process, purity is not affected by biomass type, polymer type and content. On the other hand, the recovery yield is primarily influenced by polymer type and content, with the recovery of PHB being more effective compared to that of PHB-co-HV.

## 4. Conclusions

This study validated the feasibility of a simple, economical, sustainable and easily reproducible extraction method based on 1,3-dioxolane, an environmentally friendly and safe solvent, for extracting PHB and PHB-co-HV. The proposed protocol is based on the knowledge previously acquired from similar studies, but the modifications made in terms of preparation and execution make it extremely versatile; it also adapts to less known and established cases, such as the extraction of PHB-co-HV and the use of mixed methanotrophic strains. Additionally, it does not require biomass pretreatment and only uses a thermostatic reactor, which is easier to use and more economical than other solutions reported in the literature. To the best of the authors’ knowledge, this is the first time this has been achieved, confirming the potential of 1,3-dioxolane extraction and making the process sustainable and more appealing on a large scale.

Biopolymer extraction with 1,3-dioxolane was carried out at the boiling temperature of the solvent and at atmospheric pressure, which are lower compared to the usual extraction conditions with 1,3-dioxolane. This allowed us to obtain recovery and purity yields comparable to those reported in the literature, significantly reducing contact time.

By reducing the extraction time, the purity of the polymers generally increases, but a reduction in the recovery can be observed. The results highlight that an extraction time of 30 min is sufficient to extract highly pure PHB and PHB-co-HV, with a purity of about 99% w w^−1^ and 94% w w^−1^ for PHB and PHB-co-HV from *M. hirsuta*, respectively, and 98% w w^−1^ for both polymers from the mixed methanotrophic strain. Purity is not affected by biomass type, polymer type and content. At the same time, the recovery yield is influenced by polymer type and content, with the recovery of PHB being more effective compared to that of PHB-co-HV. However, additional efforts are required to investigate the process conditions, ensuring that purity and recovery simultaneously achieve very high yields (≥97–98%).

## Figures and Tables

**Figure 1 polymers-16-01910-f001:**
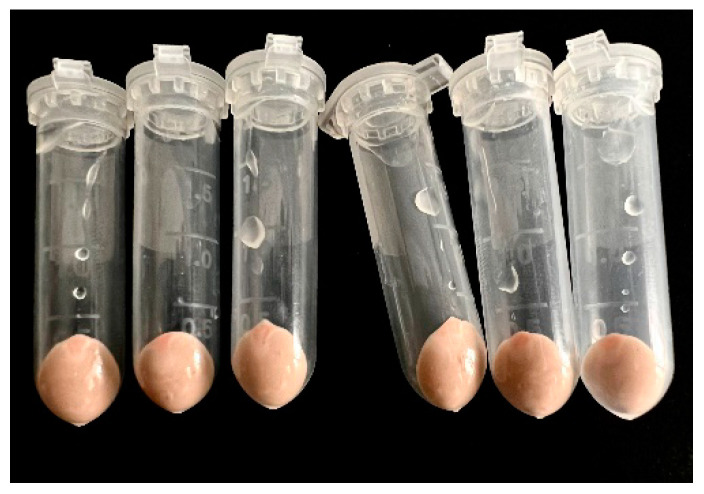
Wet biomass of *M. hirsuta* grown in nitrogen-free medium and used for PHA extraction experiments.

**Figure 2 polymers-16-01910-f002:**
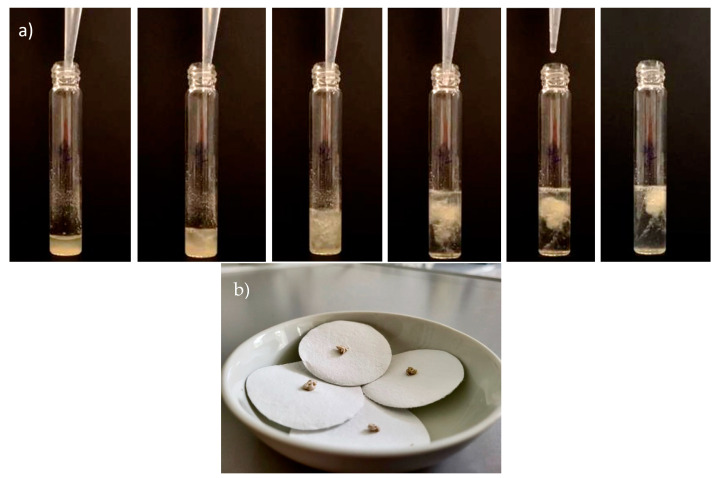
Biopolymer precipitated (**a**); biopolymer dried on filter paper overnight at room temperature (**b**).

**Figure 3 polymers-16-01910-f003:**
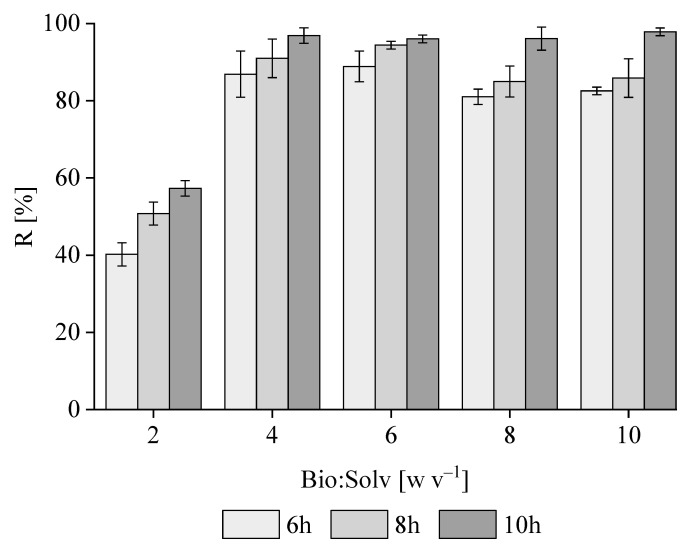
The influence of the biomass-to-solvent ratio (w v^−1^) and extraction time on the recovery of PHB from *M. hirsuta*.

**Figure 4 polymers-16-01910-f004:**
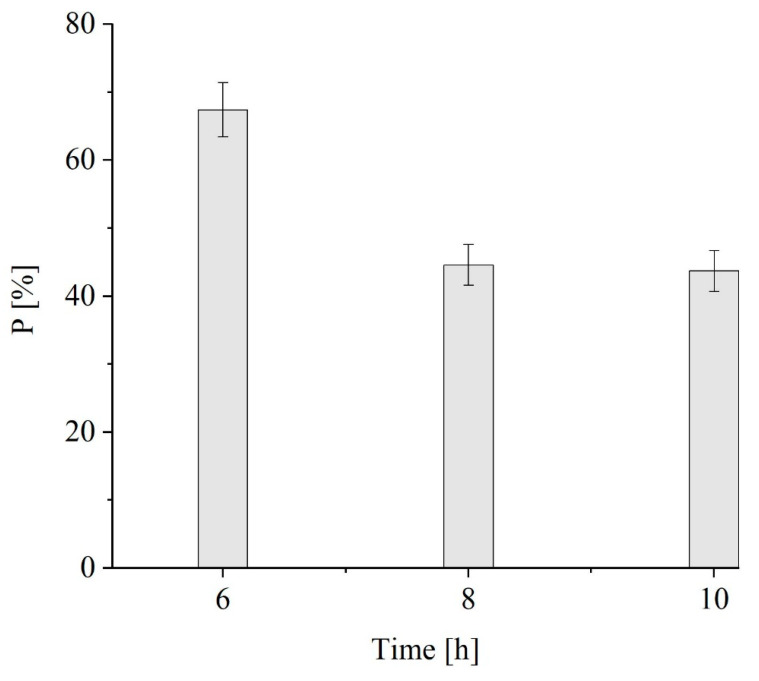
The influence of the extraction time on the average P of the extracted PHB.

**Figure 5 polymers-16-01910-f005:**
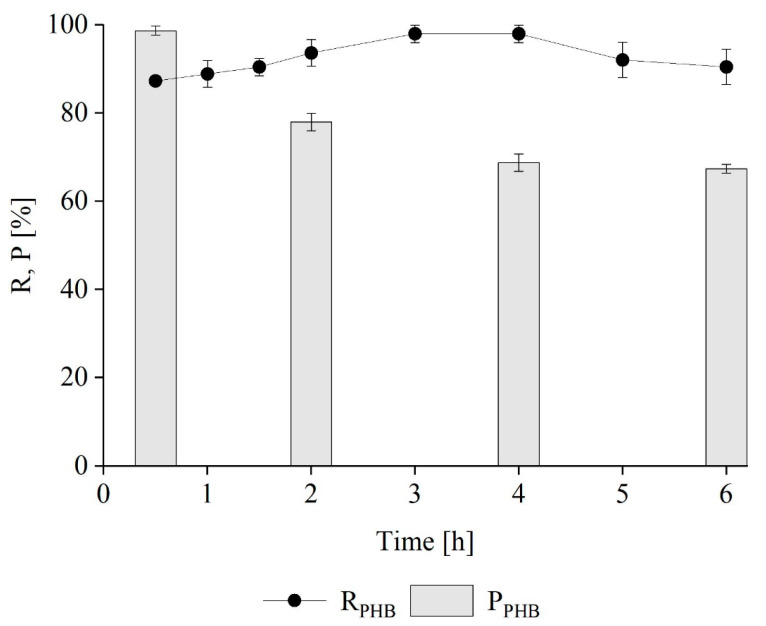
Time course of R and P during PHB extraction from *M. hirsuta*.

**Figure 6 polymers-16-01910-f006:**
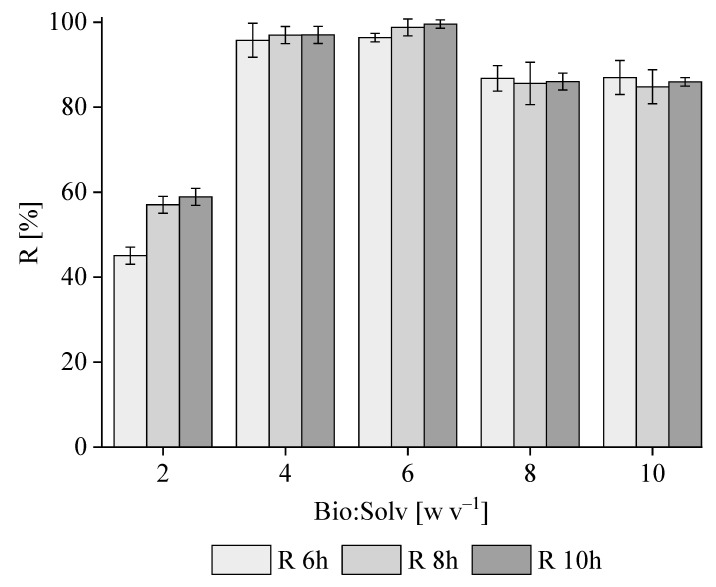
The influence of the biomass-to-solvent ratio (w v^−1^) and contact time on the recovery of PHB-co-HV from *M. hirsuta*.

**Figure 7 polymers-16-01910-f007:**
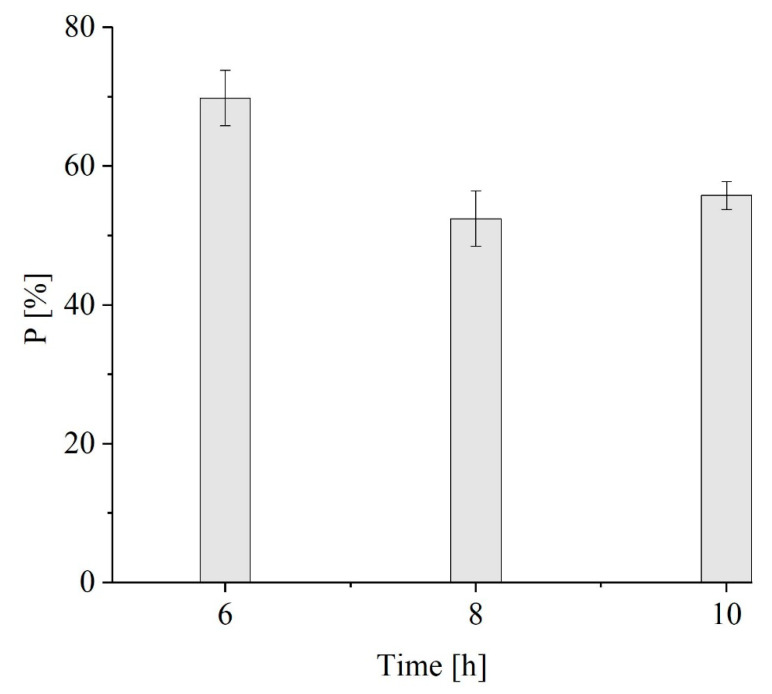
The influence of the extraction time on the average P of the extracted PHB-co-HV.

**Figure 8 polymers-16-01910-f008:**
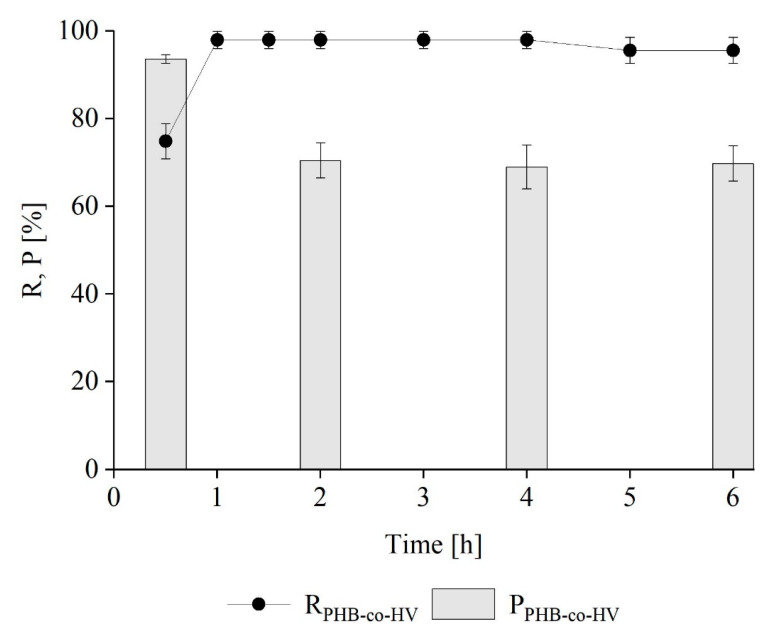
Time course of R and P during PHB-co-HV extraction from *M. hirsuta*.

**Figure 9 polymers-16-01910-f009:**
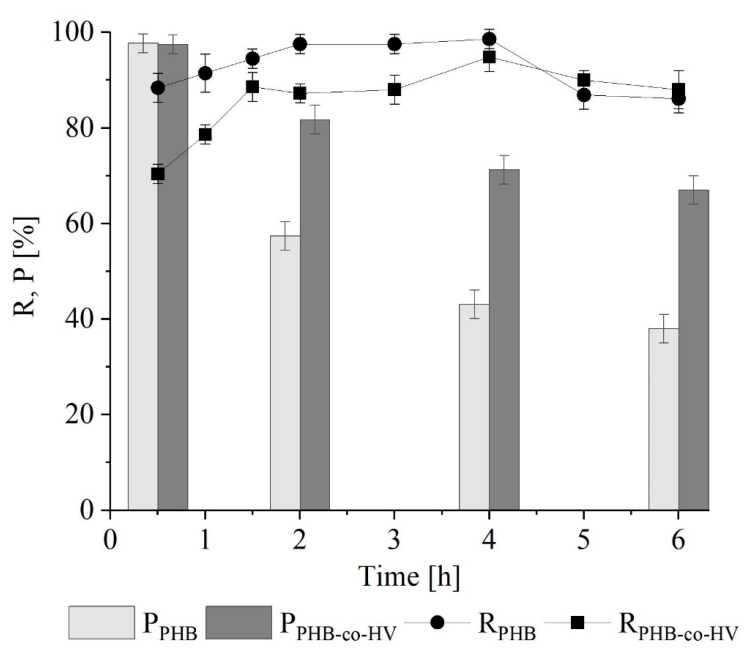
Time course of R and purity (P) during PHB and PHB-co-HV extraction from mixed methanotrophic culture.

**Table 1 polymers-16-01910-t001:** Comparison of recovery and purity yields obtained for PHB and PHB-co-HV.

Polymer	PHB		PHB-co-HV	
Biomass	*M. hirsuta*	Mixed Consortium	*M. hirsuta*	Mixed Consortium
Content of PHAs [% w w^−1^]	62	48	60	35
R [%]	87 ± 1	88.4 ± 3	75 ± 4	70 ± 4.5
P [%]	99 ± 0.9	96 ± 2	94 ± 1	97 ± 2.2

## Data Availability

The original contributions presented in the study are included in the article, further inquiries can be directed to the corresponding author.

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
