# Peer review of "Exploring 1,3-Dioxolane Extraction of Poly(3-hydroxybutyrate) and Poly(3-hydroxybutyrate-co-3-hydroxyvalerate) from Methylocystis hirsuta and Mixed Methanotrophic Strain: Effect of Biomass-to-Solvent Ratio and Extraction Time"

_polymers, 2024, doi:10.3390/polym16131910_

Round 1
Reviewer 1 Report
Comments and Suggestions for Authors
The manuscript is devoted to a novel technique of extraction of P3HB and P3HB-3HV from Methylocystis hirsute and other bacteria biomass by 1,3-dioxolane. The study is valuable in this scientific area: the problem of isolation and purification of PHAs from bacterial biomass is still a problem of high relevance. The manuscript is well organized, well documented and well written. This manuscript may be accepted. But there are some serious issues that need revision.
1. What was the P3HB and P3HB-3HV content in the Methylocystis hirsute cells? This is a very important parameter on which the efficiency of polymer extraction with any solvents depends to a great extent.
2. Data on polymer extraction parameters at different polymer contents in bacterial cells should be given.
3. Investigators do not compare the extraction parameters of polymers from bacterial biomass with 1,3-dioxolane and conventional organic solvents used for P3HB and P3HB-3HV extraction, such as chloroform and methylene chloride. How else can you understand whether this method is more effective or not?
4. How toxic this solvent is and whether more of it remains in the polymer matrix than, for example, chloroform when P3HB or P3HB-3HV is precipitated from a solution of these solvents?
Author Response
Dear Reviewer 1,
thank you for your feedback and the time spent revising the manuscript. Attached you can find the detailed response letter to your comments.

Reviewer 2 Report
Comments and Suggestions for Authors
1. Abstract should start from rationale of the study.
2. Some abbreviations are used without explaining their full word. Please do the needful.
3. There are some grammatical mistakes in the manuscript. Please revise it thoroughly.
4. What is new and novel in this study beyond the already reported literature?
5. Some results are not supported with literature. Please do the needful.
6. Please include the limitations in conclusion section.
7. Please cite some relevant latest papers of the respective Journal.
Comments on the Quality of English LanguageSome grammatical mistakes are observed in the manuscript, that should be fixed before the decision of the manuscript.
Author Response
Dear Reviewer 2, thank you for your feedback and the time spent revising the manuscript. Attached you can find the detailed response letter to your comments.

Reviewer 3 Report
Comments and Suggestions for Authors
1. The long introduction of plastic and its problem is not warranted. Just a single statement justification for bioplastic would be sufficient. Brevity in writing is expected.
2. If 1,3-dioxolane is already "...known to be least efficient in terms of final process yield, but it has many advantages, including a good quality product, compatibility with water, affordable prices, easy availability and safety", what is the relevance of this work?
3. What is NSM? We refer to it as a mineral salt medium (MSM). please check and correct it. The trace-element solution concentrate is separately prepared, and a small volume is added to the MSM solution (containing the macro-elements). Writing a composition to 7 significant digits (i.e., 0.000015) is not acceptable).
4. The biomass and Non-PHA cell biomass (NPCB) are two distinct measurements for cells containing intracellular products. Please clarify this aspect for Bio:Sol ratio in this study.
5. The statement "The frozen biomass samples, i.e. 10.2 mg for PHB extraction and 8 mg for PHB-co-HV extraction (on a dry basis), were initially thawed overnight at 4°C, and 1,3-dioxolane was added to reach biomass-to-solvent ratios (w v-1) of 2, 4, 6, 8, 10" is confusing, also specify the exact volumes rather than only ratios.
6. "performance of the extractions was evaluated in terms of R and P." this statement requires the R and P to be predefined or immediately below the statement. Again, if recovery was calculated in triplicate, why only the average value was reported? I would recommend the inclusion of replicates in statistical analysis, i.e. one- or two-way ANOVA analysis.
7. It is not clear why the author would have chosen two different time scales for PHA recovery and purity.
8. Based on Figure 3 and pertinent discussion, A two-way ANOVA analysis, followed by multiple comparison tests should be essential before you can fix the optimal biomass-to-solvent ratio and extraction time. Currently, the adopted strategy is biased by authors' preferences.
9. "Hence, the trade-off between recovery and purity yield.." You can not have a trade-off unless relative importance between the two is clearly defined, and composite desirability is mathematically formulated. This is an optimization problem. I would recommend that the author can formulate a mathematical model as a function of polymer-solvent ratio, and time.
Comments on the Quality of English LanguageBrevity in writing is missing, should have a shortened introduction.
Author Response
Dear Reviewer 3, thank you for your feedback and the time spent revising the manuscript. Attached you can find a detailed response to your comments.

Round 2
Reviewer 1 Report
Comments and Suggestions for Authors
My comments have been corrected in full, the manuscript may be published in its current form.
Reviewer 2 Report
Comments and Suggestions for Authors
Authors have addressed all comments properly.